# AMPA receptor mediated synaptic excitation drives state-dependent bursting in Purkinje neurons of zebrafish larvae

**Mohini Sengupta, Vatsala Thirumalai\***

National Centre for Biological Sciences, Bangalore, India

**Abstract** Purkinje neurons are central to cerebellar function and show membrane bistability when recorded in vitro or in vivo under anesthesia. The existence of bistability in vivo in awake animals is disputed. Here, by recording intracellularly from Purkinje neurons in unanesthetized larval zebrafish (*Danio rerio*), we unequivocally demonstrate bistability in these neurons. Tonic firing was seen in depolarized regimes and bursting at hyperpolarized membrane potentials. In addition, Purkinje neurons could switch from one state to another spontaneously or with current injection. While GABA$_A$R or NMDAR were not required for bursting, activation of AMPARs by climbing fibers (CFs) was sufficient to trigger bursts. Further, by recording Purkinje neuron membrane potential intracellularly, and motor neuron spikes extracellularly, we show that initiation of motor neuron spiking is correlated with increased incidence of CF EPSPs and membrane depolarization. Developmentally, bistability was observed soon after Purkinje neuron specification and persists at least until late larval stages.

## Introduction

Cerebellar Purkinje neurons have long fascinated neuroscientists due to their elaborate dendritic arbors (*Ramón y Cajal, 1911*), and their high rates of spontaneous activity in vitro (*Llinás and Sugimori, 1980*; *Raman and Bean, 1999*; *Han and Bell, 2003*) and in vivo (*Armstrong et al., 1979*; *Kitamura and Häusser, 2011*). Being the sole output neuron of the cerebellar cortex, the firing patterns of Purkinje neurons are functionally significant for motor co-ordination and motor learning (*Thach, 1968*; *Medina, 2011*; *Yang and Lisberger, 2014*; *Hewitt et al., 2015*). In addition, mammalian Purkinje neurons exhibit membrane bistability in vitro (*Llinás and Sugimori, 1980*; *Oldfield et al., 2010*) and in vivo under anesthesia (*Loewenstein et al., 2005*; *Schonewille et al., 2006*), transitioning between 'up' and 'down' states. The existence of bistability in Purkinje neurons of awake animals has been the subject of some debate (*Schonewille et al., 2006*; *Yartsev et al., 2009*; *Engbers et al., 2013*) not least because of the difficulty of obtaining intracellular recordings from Purkinje neurons of awake animals.

Zebrafish are a convenient model system for studying the intracellular dynamics of Purkinje neurons using whole-cell patch clamp recording in vivo in the context of simple motor behaviors. The zebrafish cerebellum has three major parts namely, the corpus cerebelli (CCe), the valvula cerebelli (Va) and the vestibulo lateral lobe which is subdivided into eminentia granularis and lobus caudalis cerebelli (*Bae et al., 2009*; *Hibi and Shimizu, 2012*). Of these only the CCe and Va show the characteristic three layered architecture (*Hashimoto and Hibi, 2012*). We have focused on Purkinje neurons in the CCe. The structure and circuitry of CCe in zebrafish are similar to that seen in electric fish and in mammals (*Meek et al., 2008*; *Hashimoto and Hibi, 2012*). As in mammals, Purkinje neurons in zebrafish also receive excitatory inputs from parallel fibers (PFs) and climbing fibers (CFs) and inhibition from stellate cells (*Figure 1A*). Purkinje neurons in zebrafish, similar to mammalian Purkinje neurons, also have large planar dendritic arbors decorated with spines (*Miyamura and Nakayasu, 2001*). However, unlike

**\*For correspondence:**
vatsala@ncbs.res.in

**Competing interests:** The authors declare that no competing interests exist.

**eLife digest** Ever wonder how you keep your balance? This is something that we learn to do as toddlers, and it involves the coordinated effort of various muscles in the body. An area at the base of the brain called the cerebellum controls this effort, and synchronizes our muscles by sending messages in the form of electrical signals along cells called Purkinje neurons. These signals consist of steady 'tonic' activity or short 'bursts' of activity. Previous studies in unconscious mammals suggest that these neurons can spontaneously switch between the two types of electrical signals.

However, it is not clear whether this switch occurs in awake animals, or how these nerve activities control muscle movements. It is technically challenging to record the voltage of single Purkinje neurons of conscious rodents, so Sengupta and Thirumalai avoided this problem by using zebrafish larvae instead. These larvae are small, transparent and lack a skull, which makes it possible to record the activity of the cerebellum using tiny glass electrodes.

The experiments show that even when the larvae are awake, the Purkinje cells produce either spontaneous bursts or tonic activity and they can readily switch between the two. The switch is controlled in part by the voltage on the neurons' surface. A positive voltage is called the 'up' state, while a negative voltage is dubbed the 'down' state. Neurons in the 'up' state produced tonic pulses, while neurons in the 'down' state produced short bursts of activity with the help of an ion channel called AMPAR.

Cells called motor neurons in the spinal cord carry the final command from the nervous system to the muscles. Sengupta and Thirumalai recorded the activity of motor neurons and Purkinje neurons at the same time. This revealed that Purkinje neurons receive a copy of the motor command that goes to the muscle and produce short bursts of electrical activity in response. This effect required AMPAR activity, and was blocked by molecules that inhibit AMPAR. Furthermore, Sengupta and Thirumalai report that the timing of these short bursts with respect to the motor command changes from one Purkinje neuron to another. Future work will investigate how the Purkinje neurons receive and process the information in the motor command.

mammals, zebrafish Purkinje neurons do not project to deeper layers but to efferent cells whose somata are present in the Purkinje cell layer or slightly ventral to it (*Bae et al., 2009*; *Takeuchi et al., 2015*). These efferent cells, also called the eurydendroid cells, receive short axons from the Purkinje neurons, send long projections outside the cerebellum (*Bae et al., 2009*; *Heap et al., 2013*; *Takeuchi et al., 2015*) and are equivalent to the mammalian deep cerebellar nuclear cells (*Figure 1A*). Zebrin II (aldolase-c, fructose bis-phosphate, a; *aldoca*) is expressed in all Purkinje neurons in zebrafish (*Miyamura and Nakayasu, 2001*; *Bae et al., 2009*), although it is expressed in only a subset of Purkinje neurons in mammals (*Doré et al., 1990*; *Brochu et al., 1990*; *Leclerc et al., 1992*; *Sillitoe et al., 2004*) and birds (*Pakan et al., 2007*; *Iwaniuk et al., 2009*; *Corfield et al., 2015*; *Vibulyaseck et al., 2015*). We use the *aldoca* promoter to fluorescently label Purkinje neurons specifically so as to target them for whole-cell patch clamp recordings (*Tanabe et al., 2010*; *Takeuchi et al., 2015*).

In zebrafish, cerebellar cells start to differentiate at three days post fertilization (dpf) and a simple layered structure forms by 5 dpf (*Bae et al., 2009*). Purkinje neurons appear at 2.8 dpf and their numbers stabilize at around 300 neurons by 5 dpf (*Bae et al., 2009*; *Hamling et al., 2015*). By this stage, PF and CF inputs to Purkinje neurons are already present (*Bae et al., 2009*; *Takeuchi et al., 2015*; *Hamling et al., 2015*). Previous studies using calcium imaging and extracellular recording have shown that Purkinje neurons are electrically active at 6–7 dpf (*Ahrens et al., 2012*; *Hsieh et al., 2014*; *Matsui et al., 2014*). Here, we undertake a thorough characterization of spontaneous activity patterns in Purkinje neurons of larval zebrafish. We show that zebrafish Purkinje neurons exhibit membrane bistability in vivo and that the state changes are dependent on AMPAR-mediated CF inputs.

## Results

### Purkinje neurons show three types of spontaneous events

We targeted Purkinje neurons for extracellular and intracellular recordings using *aldoca* promoter driven Venus expression as a marker (*Figure 1B–F*). Loose patch recordings from larval zebrafish

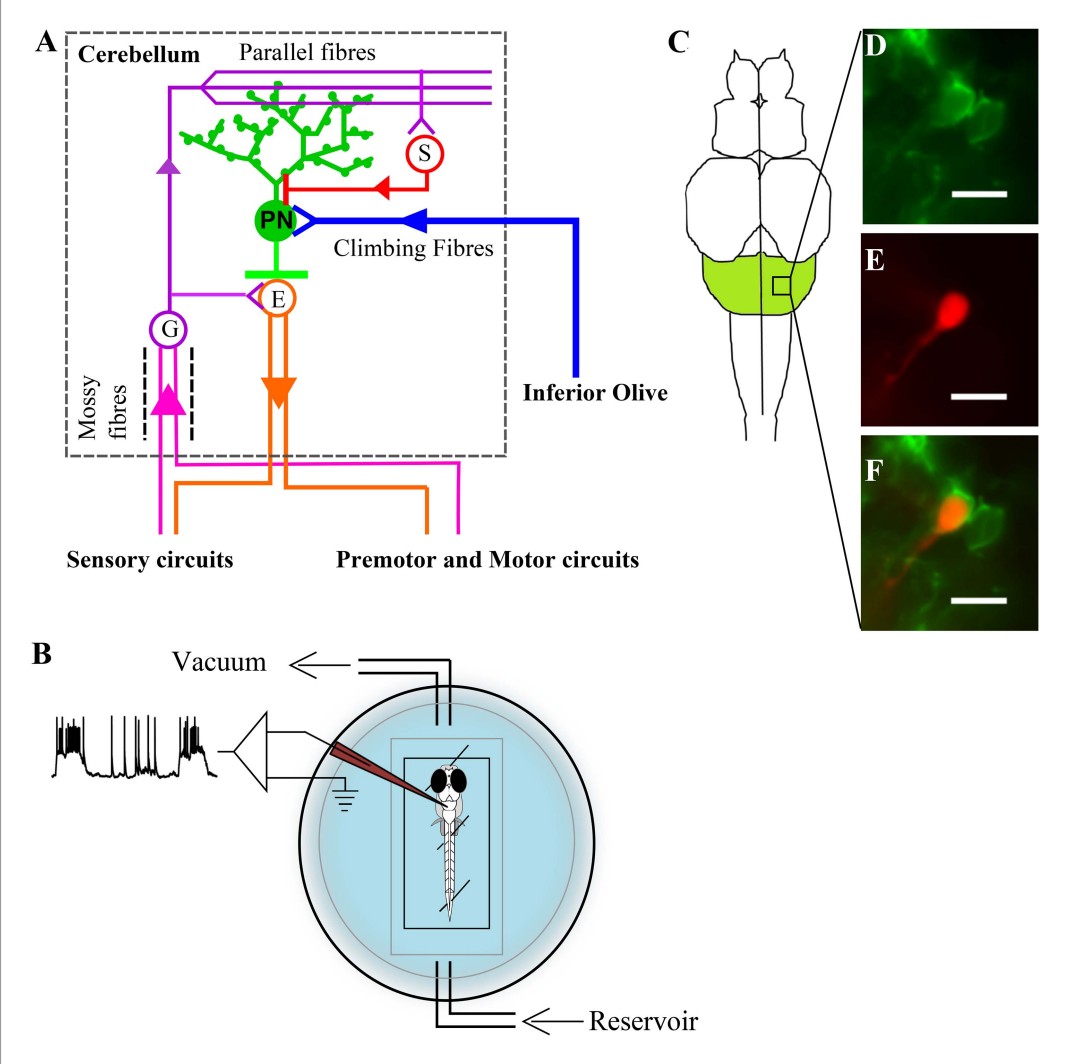

**Figure 1**. Zebrafish cerebellar circuitry and experimental preparation. (**A**) Schematic of the cerebellar circuitry in zebrafish. PN: Purkinje neuron; E: Eurydendroid cell; G: Granule cell; S: Stellate cell. (**B**) Schematic of the set up for in vivo whole cell recordings in unanesthetized zebrafish larvae. (**C**) Schematic of the zebrafish larval brain with the cerebellum in green. (**D**) Mosaic expression of *aldoca:gap43-Venus* in Purkinje neurons. (**E**) Patched cell shown filled with sulphorhodamine. (**F**) Co-localization of sulphorhodamine filled cell with membrane-targeted Venus expression. Scale bar = 10 μm.

Purkinje neurons revealed either tonically spiking (*Figure 2A*, top trace) or bursting (*Figure 2A*, bottom trace) activity. We also observed two types of events in all the cells we recorded from and these differed in amplitude and waveform (*Figure 2B*; N = 6 cells). The large amplitude events had two distinct phases typical of CF mediated responses that have been extracellularly recorded in mormyrid fish (*Alviña and Sawtell, 2014*) and in zebrafish (*Hsieh et al., 2014*). The small amplitude events resembled the simple spikes recorded earlier in the above mentioned studies.

To further investigate these events in greater detail, we performed whole cell patch clamp on these neurons. We recorded from 7 dpf larvae showing mosaic expression of *aldoca:gap43-Venus* in Purkinje neurons (*Figure 1B–F*). All cells had input resistances of above 800 MΩ (2.8 ± 0.75 GΩ, N = 13 cells) and capacitance ranging from 5.6 to 18 pF (13 ± 2.1 pF). Resting membrane potential varied from −40.5 mV to −60.2 mV (−50.3 ± 2.1 mV). As observed in the extracellular recordings, we found mainly two modes of spontaneous activity when we recorded in whole cell mode: bursting and

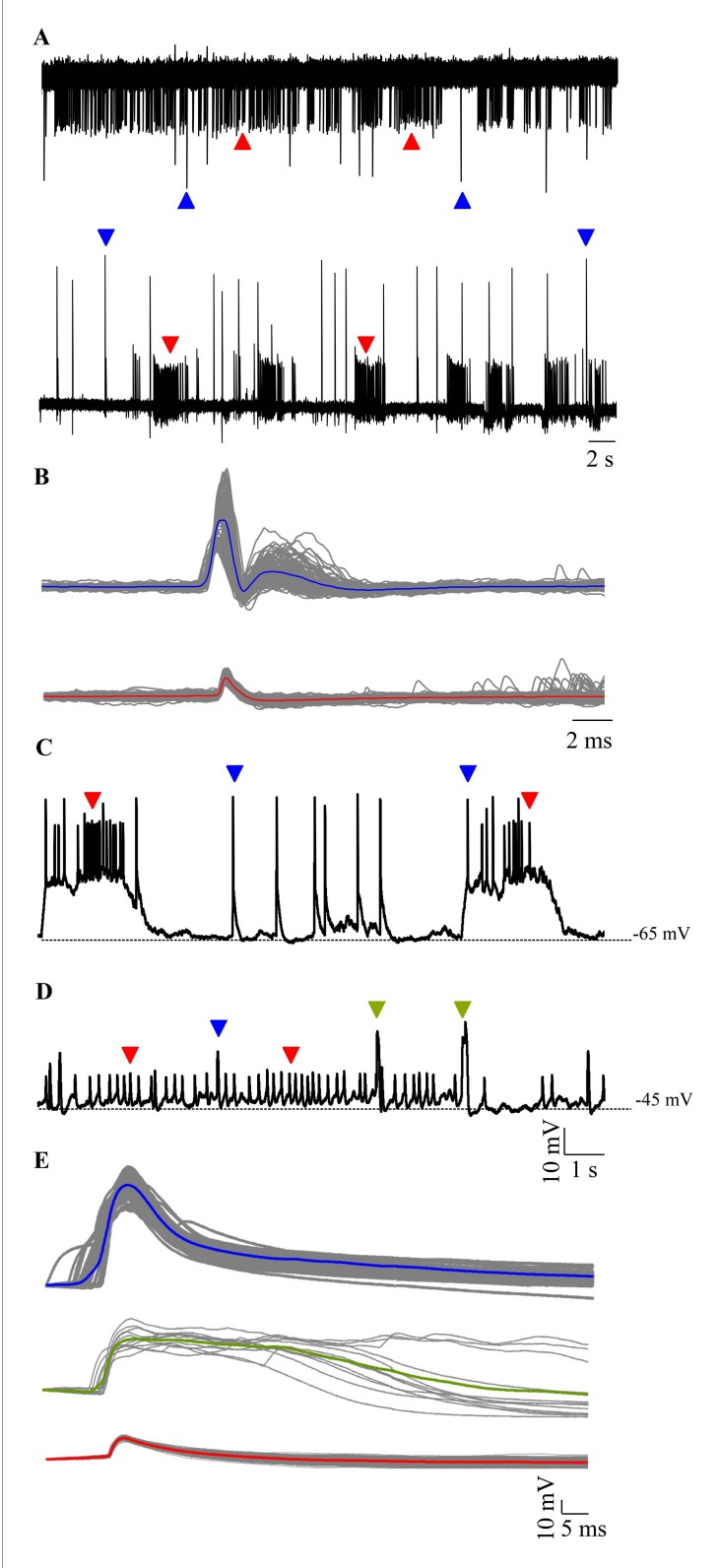

**Figure 2**. Spontaneous activity in Purkinje neurons. (**A**) Representative traces of loose patch recording from a Purkinje neuron at 7 dpf showing tonic (top trace) and bursting (bottom trace) activity patterns. Small amplitude events (red arrowheads) interspersed with large amplitude events (blue arrowheads) can be seen. (**B**) Superimposed events of large amplitude (blue) and small amplitude (red) events from one cell showing the mean in the respective

*Figure 2. continued on next page*

*Figure 2. Continued*

colour (N = 6 cells). (**C**–**E**) Intracellularly, Purkinje neurons show three types of events. Current clamp recordings of spontaneous activity at −65 mV (**C**) and −45 mV (**D**) showing large amplitude events (blue arrowheads, **C**, **D**), small amplitude narrow spikes (red arrowheads, **C**, **D**) and broad spikes (green arrowheads, **D**). (**E**) Superimposed events from one cell with mean shown in the respective colour for each type of event.

tonic (*Figure 2C,D*). 6 out of 13 cells showed bursts of activity punctuated by large amplitude events (*Figure 2C*, blue arrowheads), while the rest showed tonic spiking interspersed with broad spikes and large amplitude events (*Figure 2C,D*, green and blue arrowheads respectively). Cells in the bursting mode showed narrow attenuated spikes riding atop depolarizations (*Figure 2C*, red arrowheads). Spikes were of larger amplitude and occurred at higher frequencies during bursts than in the tonic mode (*Table 1*). Narrow attenuated spikes, broad spikes and large amplitude events had distinct peak amplitudes and kinetics (*Figure 2E* and *Table 1*; Source data in *Table 1—source data 1*). By comparison with in vitro recordings of Purkinje neurons in the central lobe of mormyrid fish (*Han and Bell, 2003*; *de Ruiter et al., 2006*), we concluded that the narrow spikes were sodium-dependent action potentials, broad spikes were calcium-mediated spikes and the large amplitude events were CF EPSPs. Nevertheless, since these are the first intracellular recordings from zebrafish Purkinje neurons, we next confirmed that this was indeed the case.

## Narrow spikes are sodium action potentials

To test whether narrow spikes are sodium action potentials, we bath applied 1 µM Tetrodotoxin (TTX) and observed that all types of spontaneous activity were abolished (*Figure 3A*; N = 3 cells). Since TTX abolishes network activity as well, we next included 5 mM QX-314, an intracellular sodium channel blocker in our patch internal solution. QX-314 abolished the ability of cells to fire action potentials even when depolarized (*Figure 3—figure supplement 1*). In the presence of QX-314, while narrow spikes were eliminated, large amplitude events could still be seen (*Figure 3B*, N = 5 cells). In a separate set of experiments, we left sodium channels intact, but blocked network activity using a cocktail of synaptic receptor antagonists. Here, we observed that the large amplitude events were eliminated. Further, the cells were depolarized (−41.3 ± 0.56 mV) and fired tonically at 38.1 ± 7.4 Hz (*Figure 3C*; N = 5 cells). These spikes are sodium action potentials as they were abolished in the presence of TTX (*Figure 3—figure supplement 2*). These experiments demonstrate that the narrow spikes are sodium action potentials and that zebrafish Purkinje neurons fire sodium spikes tonically even when isolated from the network.

## Broad spikes are dependent on voltage-dependent calcium channel activation

We frequently observed broad spikes being recruited during depolarizing current steps. The rheobase for the broad spikes (43.1 ± 7.1 pA; N = 8 cells) was consistently higher than that of sodium spikes (8.6 ± 1.2 pA; N = 7 cells; p < 0.001; Mann–Whitney; *Figure 4A*). These broad spikes could be

**Table 1**. Summary of properties of narrow spikes, broad spikes and large amplitude events observed in Purkinje neurons at 7 dpf (N = 13 cells)

|  | Narrow spikes (bursts, 1152 events) | Narrow spikes (tonic, 997 events) | Large amplitude events (758 events) | Broad spikes (79 events) |
|---|---|---|---|---|
| Peak amplitude (mV) | 15.9 ± 0.17 | 11 ± 0.09 | 51.2 ± 0.25 | 32.8 ± 0.6 |
| Frequency (Hz) | 34.8 ± 4.5 | 11.7 ± 1.1 | 1.5 ± 0.07 | 5.6 ± 3.8 |
| Full width at half max. amp (ms) | 7.7 ± 0.1 | 11.4 ± 0.1 | 15.8 ± 0.3 | 64.7 ± 5.4 |
| Rise time, 10–90% (ms) | 3.1 ± 0.1 | 3.6 ± 0.1 | 2.2 ± 0.04 | 6.5 ± 1.2 |

**Source data 1**. Amplitudes and kinetics of the three types of spontaneous events.

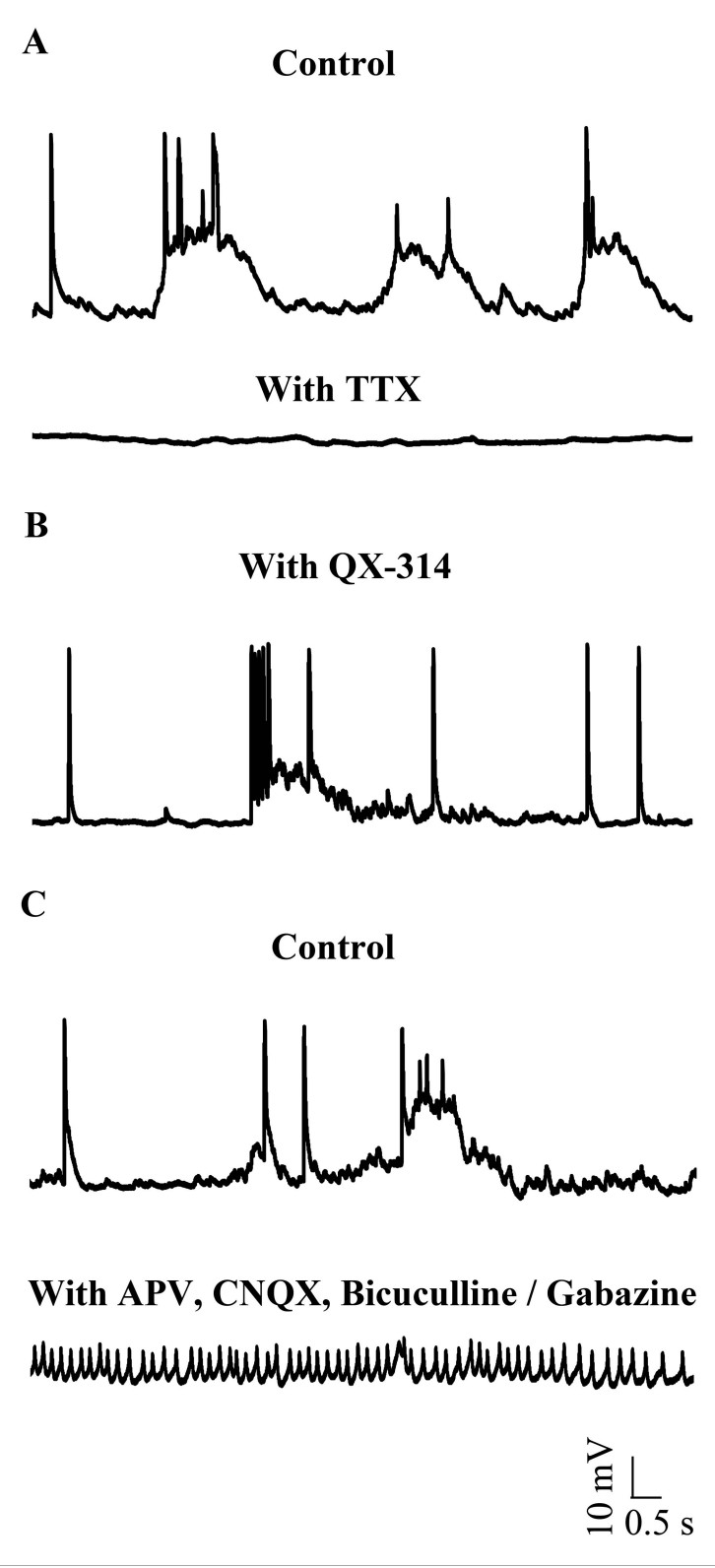

**Figure 3**. Small events are sodium dependent action potentials. (**A**) Tetrodotoxin (TTX) abolishes sodium action potentials in Purkinje neurons (N = 3 cells). Representative trace from one cell in the absence (top trace) and presence (bottom trace) of 1 µM TTX. This cell rested at −60 mV. (**B**) QX-314 also abolishes sodium action potentials (N = 5 cells). Representative trace from one cell showing absence of sodium spikes. This cell rested at −53 mV. *Figure 3. continued on next page*

*Figure 3. Continued*

(**C**) Narrow sodium action potentials occur even in the presence of synaptic receptor blockers (N = 5 cells). Representative trace from one cell in the absence (top trace) and presence (bottom trace) of APV, CNQX and Bicuculline or Gabazine. This cell rested at −42 mV.

The following figure supplements are available for figure 3:

**Figure supplement 1**. Sodium action potentials are recruited during depolarizing current injections (N = 5 cells in each condition).

**Figure supplement 2**. Narrow spikes observed in the presence of synaptic receptor blockers are abolished by TTX.

elicited by strong depolarization, even in the presence of TTX (*Figure 4B*), indicating that these events do not require voltage-gated sodium channels. We next tested whether the broad spikes required the activation of voltage-gated calcium channels by bath applying the calcium channel blocker, cadmium. Broad spikes were abolished after exposure to 200 µM Cadmium chloride (*Figure 4C*), thus indicating that these events are calcium spikes triggered when the cell is depolarized. Calcium spikes were seldom observed in cells that were in the bursting mode, but were much more prevalent in the tonic mode (*Figure 2C,D*; N = 7 cells).

## Large amplitude events are putative CF EPSPs

The large amplitude events that we observed in zebrafish Purkinje neurons appeared very similar in shape and amplitude to the all or none CF EPSPs previously recorded in vitro in Purkinje neurons in the central lobe of the cerebellum of mormyrid fish (*Han and Bell, 2003*; *de Ruiter et al., 2006*). Similar to the mormyrid CF-EPSPs, these large amplitide events were always about 50 mV in amplitude and had a sharp peak and a broad shoulder (*Figure 5A*, inset and *Table 1*). To test whether the large amplitude events are of synaptic origin, we depolarized and hyperpolarized the neuron and measured peak amplitude and inter-event intervals at various membrane potentials. As will be expected of synaptic events, we observed that the peak amplitude decreased linearly with increasing depolarization (*Figure 5A*, Pearson's r = −0.8, p < 0.001, N = 5 cells), while the inter-event interval did not change (*Figure 5B*; Pearson's r = −0.04, p = 0.33). To confirm that the large amplitude events are AMPA-receptor mediated CF synaptic inputs, we recorded in voltage clamp mode and bath applied various glutamatergic receptor antagonists. In the presence of the NMDAR blocker APV, the large amplitude events were not affected (*Figure 5—figure supplement 1*, N = 5 cells). However, the AMPAR blocker CNQX completely abolished the large amplitude events (*Figure 5C*; N = 12 cells), thus showing that these are indeed AMPAR-mediated synaptic currents. Consistent with this, these events reversed at around +12 mV (*Figure 5D*), close to AMPAR reversal potential. We next stimulated the CFs at their point of entry into the cerebellum in the presence of APV and Gabazine to eliminate NMDAR-dependent and $GABA_A$R-dependent synaptic responses. At low stimulation intensities, no response was seen (*Figure 5E*, flat line). As the stimulus amplitude was gradually increased, large amplitude EPSCs similar in size to the spontaneously occurring large amplitude events were seen (*Figure 5E*). These were all-or-none in that no EPSCs of intermediate amplitudes were seen. When CNQX was added to the bath, the all-or-none CF EPSCs were completely abolished, confirming that the large amplitude events are AMPAR mediated all-or-none CF EPSPs.

In addition to the CF EPSCs, we also observed small amplitude synaptic currents (*Figure 5—figure supplement 2*, panel A, green arrowheads), which were also abolished by CNQX, suggesting that these are putative PF mediated synaptic currents. CF and PF EPSCs were distinct in their amplitudes, with CF EPSCs consistently crossing 200 pA in peak amplitude while the PF EPSCs were usually smaller than 100 pA (*Figure 5—figure supplement 2*, panel B; p < 0.0001, Mann–Whitney test). We computed the coeffecient of variation (CV) of peak amplitudes of CF and PF EPSCs and found that the PF EPSCs had consistently larger CVs across cells (*Figure 5—figure supplement 2*, panel C; N = 6 cells; p = 0.0087, Mann–Whitney).

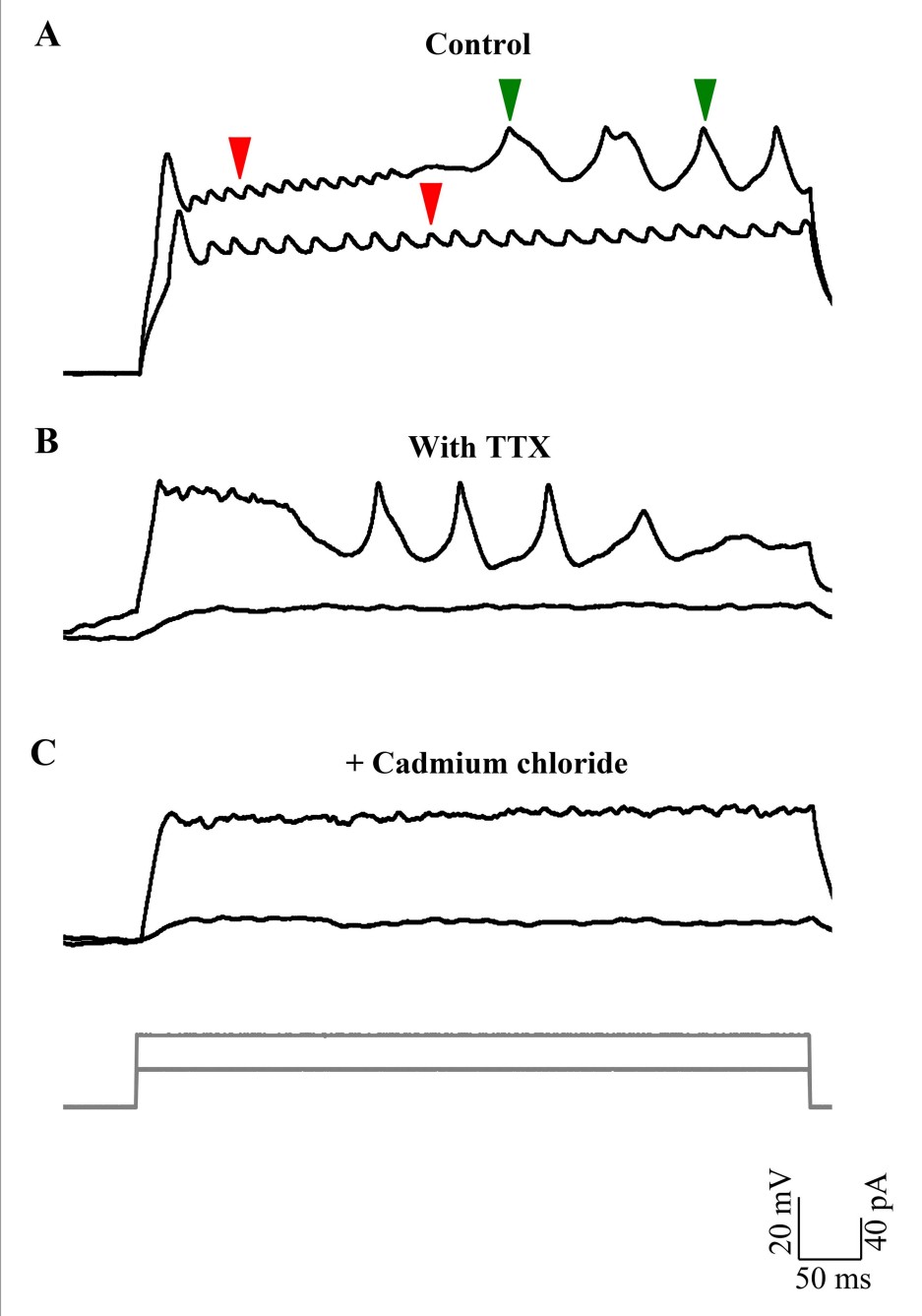

**Figure 4**. Broad events are calcium spikes. (**A**) Calcium spikes have higher rheobase than sodium spikes (N = 8 cells). Representative trace of current clamp recordings of cellular response from one cell to current injections (grey, bottom panel) showing calcium spikes (green arrowheads) being recruited at higher level of depolarization than sodium spikes (red arrowheads). (**B**) Representative traces of current clamp recordings from another cell in TTX to the same current injection protocol as in **A**. (**C**) Responses shown by the same cell as in B after 200 µM cadmium chloride was added to the bath (N = 4 cells).

## The same neuron can toggle between bursting and tonic states

Our data thus far indicated that Purkinje neurons when recorded in vivo showed bursting or tonic activity and that the activity consisted of sodium and calcium spikes as well as AMPAR-mediated all-or-none CF EPSPs. We next wanted to investigate if there were two distinct populations of Purkinje

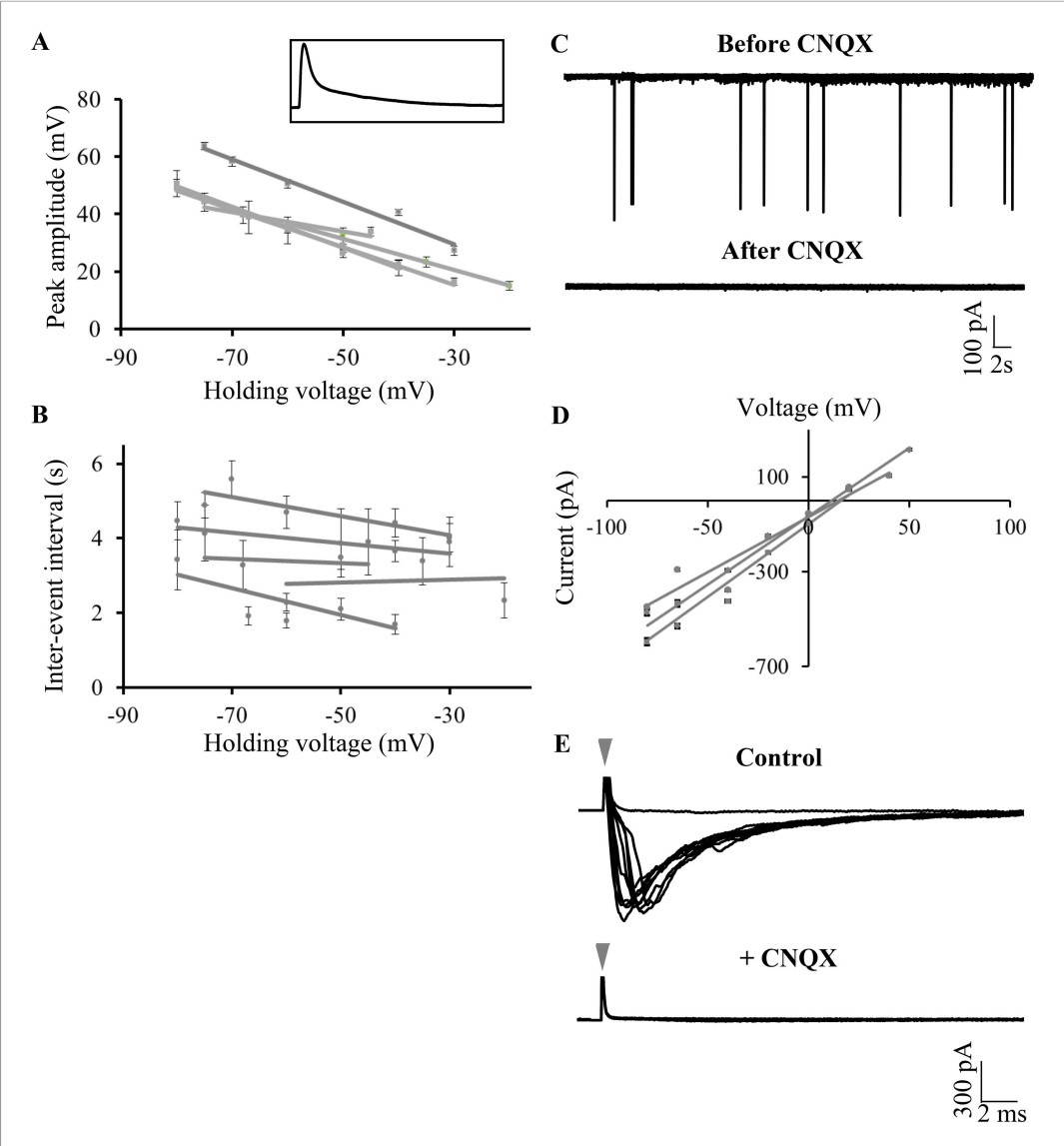

**Figure 5**. Large amplitude events are climbing fiber (CF) EPSPs mediated by AMPARs. (**A**) Mean peak amplitude of large amplitude events as a function of the holding potential. Inset: Expanded trace of a single large amplitude event to illustrate the slow kinetics and large amplitude. Inset x-axis: 400 ms; y-axis: 47 mV. (**B**) Mean inter-event interval as a function of holding potential. Error bars indicate standard error of mean in **A** and **B** (N = 5 cells). (**C**) Representative trace of a Purkinje neuron recorded in voltage clamp mode at −65 mV before and after application of CNQX. (**D**) Current-voltage relation of CF EPSCs (N = 3 cells). (**E**) Representative trace showing all-or-none EPSCs upon stimulation of CFs in the presence of APV and Gabazine (top trace; N = 7 cells). Stimulation at 500 µA resulted in either transmission failure (flat line) or EPSCs of similar amplitudes. In the same cell, all-or-none EPSCs were abolished by the addition of CNQX (bottom trace; N = 5 cells). Grey arrowhead shows stimulus artifact.

The following figure supplements are available for figure 5:

**Figure supplement 1**. Large amplitude events do not require activation of NMDA receptors.

**Figure supplement 2**. CF and PF glutamatergic synaptic inputs are clearly distinguishable.

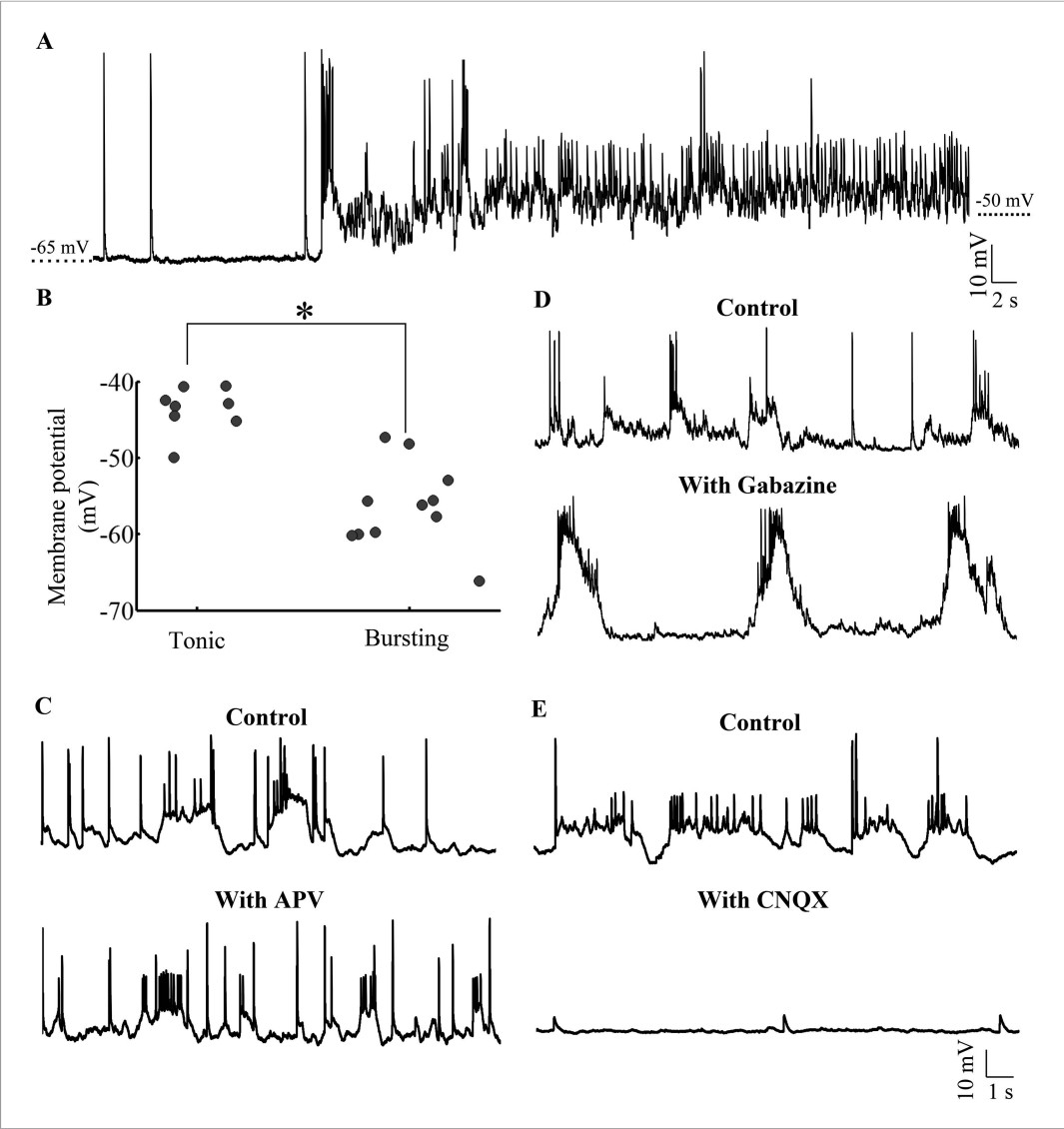

**Figure 6**. Purkinje neurons toggle between bursting and tonic states as a function of membrane potential and AMPAR-mediated synaptic input. (**A**) Representative trace of a bursting cell (bursts not shown) which rested at −65 mV spontaneously depolarizing and spiking tonically. (**B**) Scatter plot of membrane potential of cells in tonic vs bursting modes showing that tonic modes occur at more depolarized potentials than bursting mode (N = 8 cells for tonic mode and 11 cells for bursting mode). (**C**) Representative trace of a cell in the absence (top trace) and presence (bottom trace) of APV. This cell rested at −58 mV (N = 6 cells). (**D**) Representative trace of a cell in the absence (top trace) and presence (bottom trace) of Gabazine. This cell rested at −57 mV (N = 10 cells). (**E**) Representative trace of a cell in the absence (top trace) and presence (bottom trace) of CNQX. This cell rested at −60 mV (N = 12 cells).

neurons or whether the same neurons can toggle between these two states. As mentioned above, blocking synaptic receptors with APV, CNQX and Bicuculline/Gabazine eliminates bursting and causes the neurons to show tonic spiking (*Figure 3C*; N = 5 cells). We also found that Purkinje neurons could spontaneously toggle between the bursting and tonic states (*Figure 6A*). 3 out of 8 tonic cells spontaneously hyperpolarized during the course of our recording and entered the bursting state. Conversely, 4 out of 11 cells spontaneously depolarized and toggled from the bursting to tonically firing state. We examined the basal membrane potential in bursting and tonic cells and found that tonic cells were significantly more depolarized (−43.7 ± 1.1 mV, N = 8 cells) compared to bursting cells (−56.3 ± 1.6 mV; N = 11 cells; p < 0.001; Mann–Whitney; *Figure 6B*). In addition, 4 of 4 bursting

cells could be made to fire tonically by depolarizing them and 4 of 4 tonic cells could generate bursts when they were hyperpolarized with negative current. When we bath applied CNQX, APV and Bicuculline, tonic cells lost the ability to generate bursts upon hyperpolarization and became silent (N = 5 cells). To understand which synaptic receptors were involved in triggering bursts, we bath applied Gabazine, APV or CNQX individually. In the presence of 40 µM APV (N = 5 cells) Purkinje neurons were still able to generate bursts (*Figure 6C*). In the presence of 10 µM Gabazine (N = 10 cells), bursts, though of altered shape and amplitude, could still be seen (*Figure 6D*). However, 20 µM CNQX completely abolished bursting behavior (*Figure 6E*; N = 12 cells), indicating that AMPAR and not GABA$_A$R or NMDAR were responsible for triggering bursts in the hyperpolarized state. From these experiments, we conclude that Purkinje neurons have two stable states. Tonic spiking occurs in the 'up' state and when the neurons are in the 'down' state, AMPAR-mediated synaptic inputs can toggle the cell to 'up' states thus generating bursts.

## Toggling to 'up' states is triggered by CF-EPSPs

The CF to Purkinje neuron synapse is one of the strongest excitatory synapses in the CNS. We hypothesized that CF EPSPs can be the source of AMPAR-mediated strong excitation that toggles the Purkinje neuron to 'up' states. If this were true, burst onsets must be highly correlated with CF-EPSPs. Indeed, we observed that burst initiation was usually marked by the presence of CF-EPSPs (*Figure 7A*, blue arrowheads). When the burst initiation was calculated from the first sodium spike (*Figure 7A*, red arrowhead), we observed a significant increase in the number of CF-EPSPs from an average of 32.34 events per 200 ms bin to 92 events in a 200 ms bin immediately preceding the spike (*Figure 7B*; N = 5 cells; p < 0.001; Chi-square test). Next, we stimulated CF in the presence of Gabazine and APV and observed that CF activation triggered bursting in Purkinje neurons without fail (*Figure 7C*; N = 5 cells). Addition of CNQX abolishes the ability of CF stimulation to trigger CF-EPSPs and to trigger bursts (*Figure 7D*; N = 5 cells). These data argue that AMPAR-mediated CF-EPSPs are sufficient to trigger bursting in Purkinje neurons when the neurons are in the hyperpolarized state.

## Bursts are triggered during motor episodes

To determine the functional relevance of toggling between states, we recorded intracellularly from Purkinje neurons while simultaneously recording fictive motor patterns. We found that motor episode initiation was usually accompanied by the initiation of bursting in the Purkinje neuron (*Figure 8A*; N = 5 cells). Since bursting was triggered by CF-EPSPs (*Figure 7*), we first sought to determine whether motor episode initiation was accompanied by increased incidence of CF-EPSPs. *Figure 8B* shows a raster plot of CF EPSPs with respect to motor episode initiation across 43 motor episodes and 5 Purkinje neurons. The duration of each motor episode is shown by the length of the green-shaded box. CF-EPSPs tended to be clustered around the initiation of motor episodes (*Figure 8B* and 0 on x-axis). The number of CF-EPSPs occurring within a 1 s window after motor episode initiation (169) was significantly higher than average (40, p < 0.001, Chi-square test; *Figure 8C*). To determine whether the increased incidence of CF-EPSPs in this window, resulted in increased bursting in the Purkinje neuron, we plotted the membrane potential of Purkinje neurons within a 1 s window around the start of motor episodes (*Figure 8D*). In the cell shown in *Figure 8D*, CFs triggered depolarization within 500 ms after the initiation of motor episodes in 80.4% trials, while in 17% trials, the CF-triggered depolarization occurred within 500 ms before motor onset. In 3% trials, the cell did not show significant change in membrane potential in this window. We next compared motor-episode related bursting across the 5 Purkinje neurons we recorded from by computing average membrane potential in a 1 s window around the start of motor episodes (*Figure 8E*). The membrane potential at +0.5 s was significantly higher by 17 ± 0.93 mV (N = 107 trials, p < 0.001) than that at –0.5 s, when motor episodes were aligned at 0 s (*Figure 8D,E*). However, depolarization latency and reliability were variable across the cells we recorded from. We found that motor episode-related, CF-triggered bursting was evoked in more than 80% trials in some neurons (cells 3, 4, and 5) while it was relatively less reliable in others (77% for cell 1 and 50% for cell 2). Additionally, the average latency from motor episode onset to depolarization onset was also variable from one motor episode to another within the same cell (*Figure 8D*) and from one cell to the other (*Figure 8E*). Taken together, these data indicate that the initiation of motor episodes is accompanied by an increased incidence of CF-EPSPs in the Purkinje neuron, which toggle it to the 'up' state. However, individual Purkinje neurons exhibit variability in how they represent the time of initiation of motor episodes.

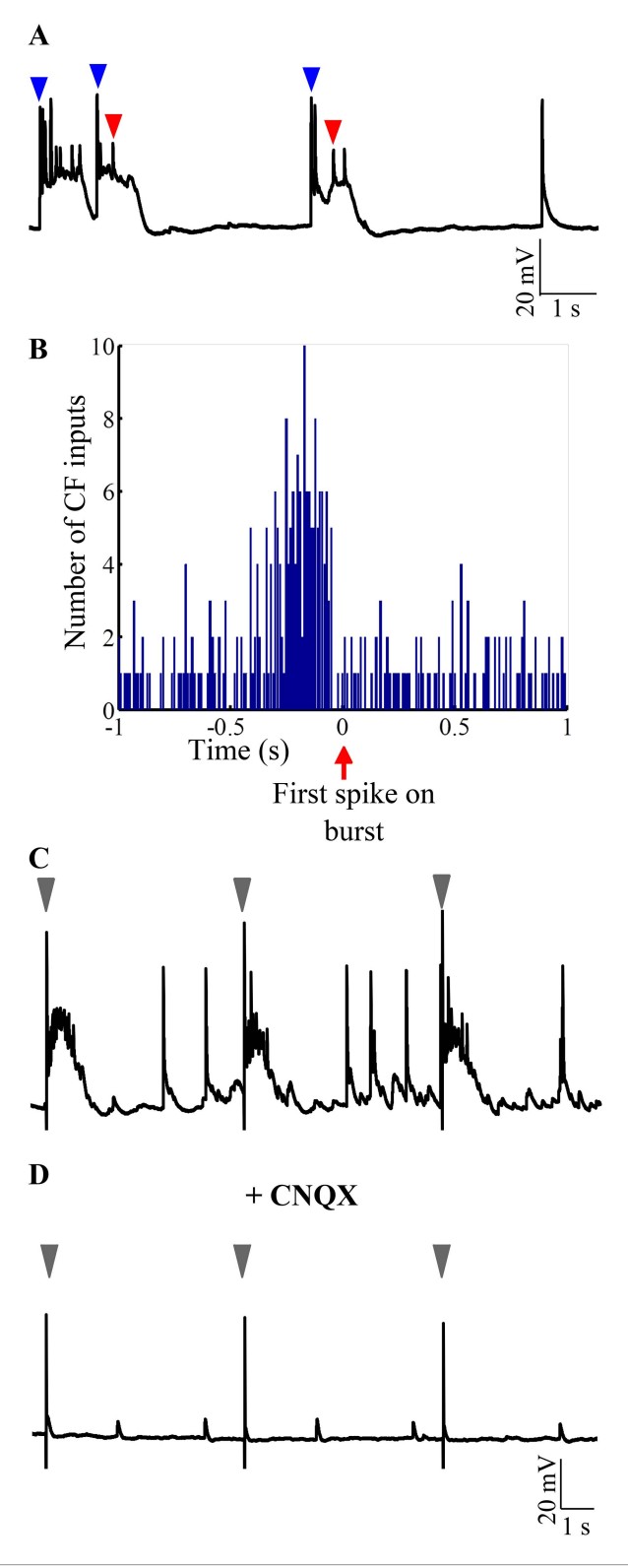

**Figure 7**. Bursting is triggered by AMPAR-mediated olivary synaptic inputs. (**A**) Representative trace showing CF EPSPs (blue arrowheads) occurring near burst onset as defined by the first sodium spike (red arrowhead). (**B**) Peri-event time histogram showing CF EPSPs clustered before the start of bursts. For every burst onset as defined by the first sodium spike, the number of CF EPSPs within a 10 s window was calculated in bins of 10 ms. This was
*Figure 7. continued on next page*

*Figure 7. Continued*

repeated for five cells and the results pooled. Only a 2 s window around burst onset is shown for greater clarity. (**C**) Representative trace showing that stimulation of CF triggers bursts. The stimulus intensity was set in voltage clamp mode at a value that did not yield any failures (N = 5 cells). (**D**) Same cell after CNQX was added to the bath solution. Grey arrowheads indicate stimulus artifacts in **C** and **D**.

## Bistability in Purkinje neurons develops early

We next investigated whether the bistable behavior that we report here can be observed in early larval stages. We recorded at 4 dpf, which is the earliest time point when Purkinje neurons can be unequivocally identified. At this stage, the spontaneous activity was indistinguishable from that seen at 7 dpf. Both bursting and tonic Purkinje neurons were present (N = 5 cells in tonic mode, 2 cells in bursting mode). CF-EPSPs could be identified, and narrow sodium spikes as well as broad calcium spikes were also present (*Figure 9A*, top trace). At 19 dpf, the latest stage we recorded from, again we were able to detect CF-EPSPs, sodium spikes and calcium spikes. Of 10 cells recorded at 19 dpf, 7 were tonically firing and 3 were bursting (*Figure 9A*, bottom trace). As in 7 dpf larvae, both at 4 dpf and at 19 dpf, tonic and bursting cells could be mode-switched by hyperpolarization and depolarization respectively. However, one key point of difference was that at 19 dpf, we observed fewer CF-EPSPs as compared to 4 dpf or 7 dpf. To determine if the number of CF inputs is reduced between 7 dpf and 19 dpf, we recorded CF-EPSCs in voltage clamp mode at 4 dpf, 7 dpf and 19 dpf. We found that the peak amplitude and inter-event interval of CF-EPSCs increased significantly between 7 dpf and 19 dpf (*Figure 9B–D*, p < 0.0001, Kruskal–Wallis), thus confirming that as the larva matures, Purkinje neurons receive fewer but stronger CF EPSCs. The coefficient of variation of CF-EPSC peak amplitude was 0.6 at 4 dpf, 0.5 at 7 dpf and 0.4 at 19 dpf, suggesting a gradual pruning of CF inputs from 4 dpf to 19 dpf.

## Discussion

We have shown that Purkinje neurons, in vivo, in the absence of anesthesia exhibit membrane bistability, with tonic firing during 'up' states and burst firing during 'down' states. Burst firing happens when the membrane is transiently toggled to 'up' states and these transitions to 'up' states can be driven by AMPAR mediated excitation. We have shown that CF inputs are the source for the strong AMPAR excitation required for these state transitions. Furthermore, initiation of motor episodes is highly correlated with higher incidence of CF EPSPs and toggling to 'up' states. Our data suggest that Purkinje neurons may utilize CF-driven bistability to create a distributed representation of the timing of motor events. In sum, we show functionally relevant bistability in Purkinje neurons that is driven by olivary input.

## Larval zebrafish as a model system to study the cerebellum and motor learning

We have recorded from Purkinje neurons of larval zebrafish from 4 dpf to 19 dpf. This preparation allows us to study intracellular membrane potential dynamics of Purkinje neurons in vivo in the context of fictive motor output. However, because it is a larval animal, it is possible that the bistability that we observe is the property of a developing network and that it may not be present in the mature cerebellum of the adult. Nevertheless, other factors suggest that the cerebellum is functionally mature at least by 7 dpf and that the bistability observed is integral to this function. Purkinje neurons are born at ~3 dpf (*Bae et al., 2009*; *Hamling et al., 2015*) and are electrically active at the earliest stage we recorded from, that is, 4 dpf. The cerebellar layers and synaptic contacts onto Purkinje neurons are observed as early as 5 dpf (*Bae et al., 2009*). Calcium imaging in 6–7 dpf larvae has revealed Purkinje neurons to be active during optomotor and optokinetic responses (*Ahrens et al., 2012*; *Matsui et al., 2014*). Optogenetic silencing of Purkinje neurons interfered with optomotor and optokinetic responses in larval zebrafish of this age (*Matsui et al., 2014*). The cerebellum has been shown to be involved in associative learning in adult vertebrates such as goldfish, rodents, rabbits and non-human primates (*Thompson and Steinmetz, 2009*; *Gómez et al., 2010*). Ablation of the entire CCe in 6–8 dpf zebrafish larvae resulted in deficits in associative learning in a classical conditioning paradigm (*Aizenberg and Schuman, 2011*). These studies provide support to the idea that the functioning of

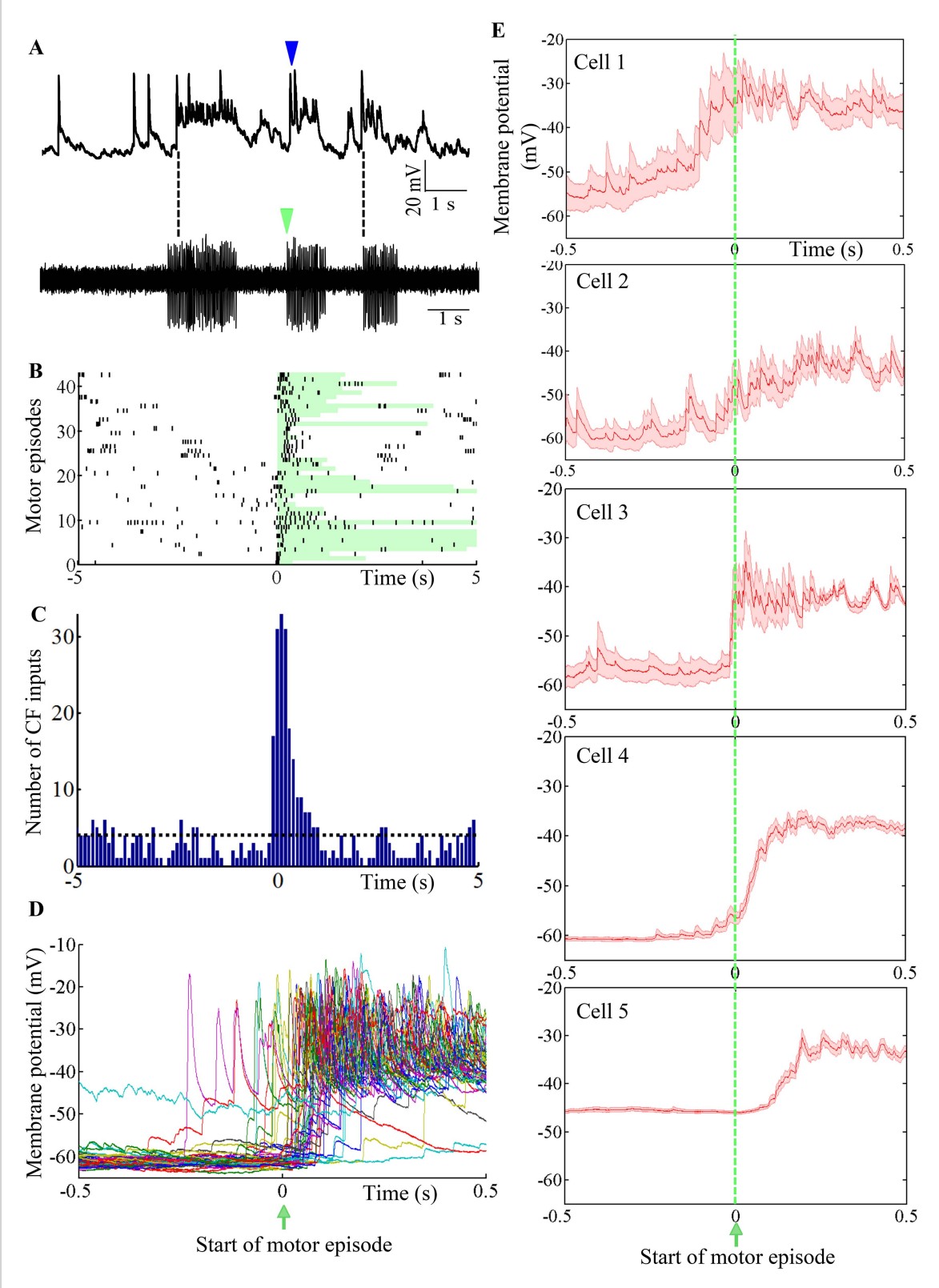

**Figure 8**. Purkinje neuron bursts are timed to occur with motor neuron bursts. (**A**) Representative trace showing intracellular recording from a Purkinje neuron (top trace) and suction electrode recording from axial myotomes (bottom trace). Purkinje neuron bursts (blue arrowhead) occur together with initiation of motor episodes (green arrowheads). (**B**) Raster plot of CF inputs (black bars) in a 10 s window around the start of motor episodes (0 on x-axis).
*Figure 8. continued on next page*

*Figure 8. Continued*

The duration of each motor episode is shown as a pale green bar. The plot shows data from 43 motor episodes arranged on the y-axis. CF-EPSPs from 5 Purkinje neurons were plotted. (**C**) Peri-event time histogram showing CF inputs clustered at the beginning of the motor episode (0 on x-axis). For every motor episode, the number of CF inputs within a 10 s window was calculated in bins of 100 ms. This was repeated for five cells and the results pooled. The dashed line indicates the average CF number per bin. (**D**) Voltage traces from a Purkinje neuron in a 1 s window around the start of motor episodes (0 on x-axis). 38 traces have been aligned and overlaid by placing the respective motor episode initiation at 0 s. (**E**) Same plot as in **D** for each of 5 Purkinje neurons. However, average membrane potential (red line) and standard error of mean (pink shaded region) are shown instead. Cell 4 is the same as the one shown in **D**.

the larval cerebellum is analogous to its function in the adult and that the activity patterns we recorded are functionally relevant in the larva and in the adult.

## Evolutionary conservation of cerebellar circuitry and physiology

Here, we have recorded intracellularly from Purkinje neurons of the zebrafish for the first time and we observe several shared as well as unique features in its physiology in comparison to that reported in mammals. The cerebellum or 'little brain' is one of the oldest parts of the vertebrate brain and is present in all vertebrates (*Bell, 2002*). Among gnathostomes, the circuitry of the cerebellum is remarkably conserved with most major cell types being present in the cerebellums of cartilaginous fish, bony fish, and tetrapods (*Nieuwenhuys et al., 2014*). Although zebrafish and mammals diverged from a common ancestor 450 million years ago, structurally, the cerebellum of zebrafish shows a high degree of similarity with mammalian cerebellar circuits. Similar to mammals, zebrafish have a layered cerebellum with molecular, ganglionic and granule cell layers. Mossy fibers innervate granule cells, which relay inputs to Purkinje neurons via PFs arranged orthogonal to their dendritic arbors. Zebrafish Purkinje neurons are also spiny with elaborate dendritic arbors (*Bae et al., 2009*; *Takeuchi et al., 2015*).

That said, several key points of structural differences have also been reported. In contrast to mammals, zebrafish do not have deep cerebellar nuclei and the efferent cells are placed proximal to Purkinje neurons within the ganglionic layer; CFs make synaptic contacts on the somata and proximal dendrites of Purkinje neurons and do not ramify extensively within the molecular layer; and basket cells have so far not been identified in zebrafish (*Bae et al., 2009*; *Takeuchi et al., 2015*). In addition, in zebrafish larvae, the rostromedial cerebellum corresponds to spinocerebellum and the caudolateral regions correspond to vestibulocerebellum; the cerebrocerebellar hemispheres are entirely absent (*Matsui et al., 2014*).

We now show that functionally also, Purkinje neurons of zebrafish have shared and unique physiological properties in comparison to mammals. Unlike mammalian Purkinje neurons, zebrafish larval Purkinje neurons do not exhibit complex spikes. Instead, they show large amplitude all-or-none CF EPSPs and broad calcium spikes. In spite of such major differences, we also identified several similarities that have been evolutionarily conserved. Similar to mammalian neurons, spontaneous firing appears to be an intrinsic property of zebrafish Purkinje neurons as these neurons continue to fire at high rates even when fast synaptic transmission is blocked, as has been shown in rodent Purkinje neurons in cerebellar slices (*Häusser and Clark, 1997*). Secondly, like in mammalian Purkinje neurons, two kinds of glutamatergic excitation are observed: strong excitation from CFs and weak excitation from PFs, both mediated by AMPA receptors (*Figure 5—figure supplement 2*). Lastly, we demonstrated bistability in Purkinje neurons of zebrafish larvae. Though bistability in Purkinje neurons has been reported in vitro in amniotes (*Llinás and Sugimori, 1980*; *Oldfield et al., 2010*) and in vivo under anesthesia in rodents (*Loewenstein et al., 2005*; *Schonewille et al., 2006*), there has been disagreement on whether it occurs in vivo under 'awake' conditions (*Schonewille et al., 2006*; *Yartsev et al., 2009*; *Engbers et al., 2013*). The zebrafish larval preparation allows us to perform intracellular recordings of membrane potential in the absence of anesthesia. Under these conditions, we observed membrane bistability and toggling between 'up' and 'down' states. Based on the conservation of other structural and functional properties of the cerebellum, it seems reasonable to conclude that bistability is a conserved feature of Purkinje neurons from fish to mammals.

## Mechanisms of toggling between tonic and bursting states

Our experiments indicate that in vivo, Purkinje neurons can be found either in the tonic or bursting mode and that they spontaneously toggle from one mode to another. When Purkinje neurons are in

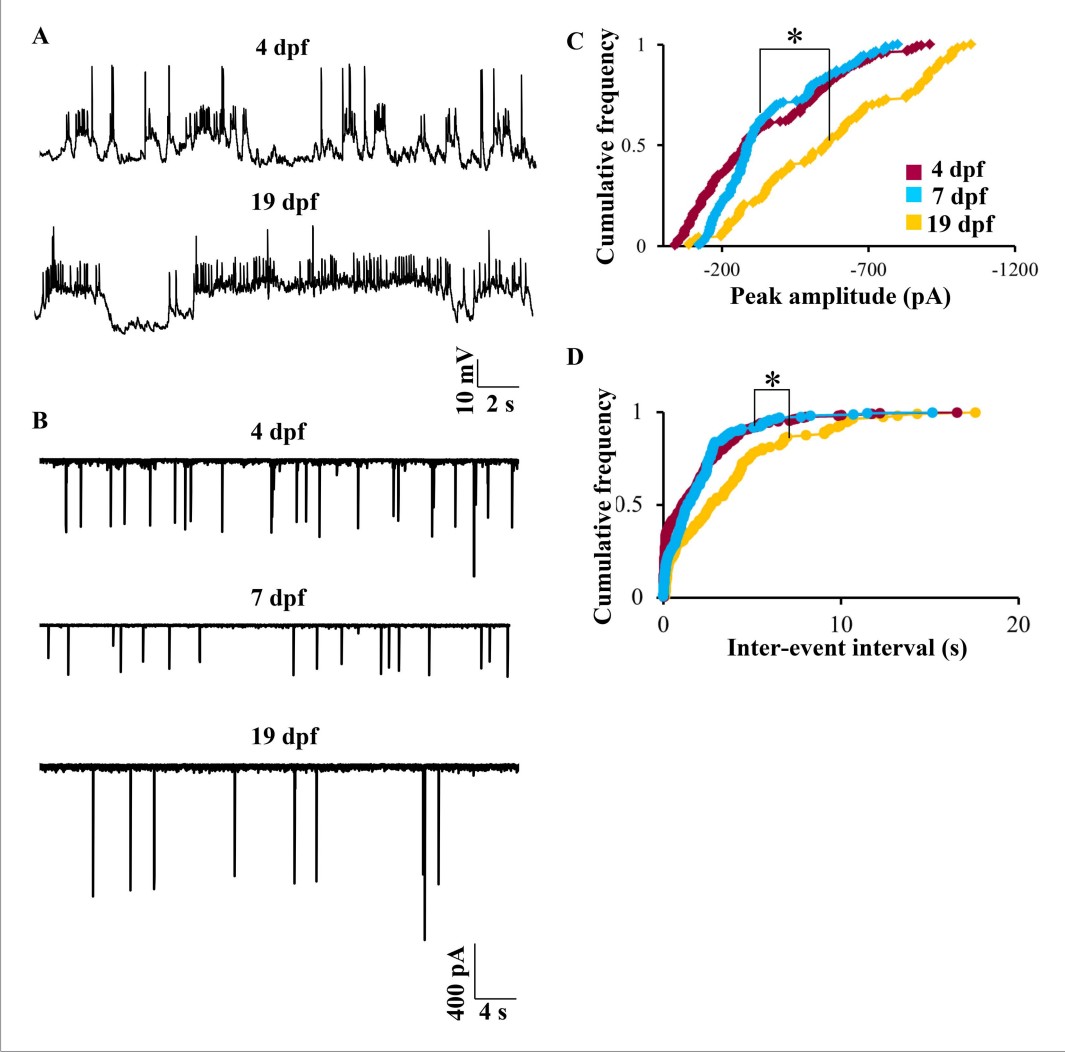

**Figure 9**. Bistability in Purkinje neurons appears soon after they are specified, but CF EPSC frequency is developmentally regulated. (**A**) Representative trace of bursting at 4 dpf (top trace) and at 19 dpf (bottom trace). (**B**) Representative traces of CF inputs at three developmental stages: 4, 7 and 19 dpf. (**C**) Cumulative frequency of the peak amplitudes of CF EPSCs at −65 mV in the three stages of development. (**D**) Cumulative frequency of inter-event interval in the three stages. The three distributions were compared using Kruskal Wallis test with a significance level of 5% followed by Tukeys post hoc analysis for pairwise comparison of means. N = 9 cells at 4 dpf; 6 cells at 7 dpf and 10 cells at 19 dpf.

the 'down' state, strong AMPAR-mediated excitation from the CFs is sufficient to trigger bursts. In zebrafish, GABAergic inhibition or NMDAR-dependent excitation seem not to be critical for generating bursts. Both a hyperpolarized membrane potential and AMPAR-mediated CF-activation were required for triggering bursts, suggesting that CF-EPSPs activate voltage-dependent conductances in the Purkinje neuron to trigger bursts. Further experiments are needed to tease out the ionic basis of the bursts triggered by CF-EPSPs.

## Ontogeny of Purkinje neuron spontaneous activity patterns

We report here that the three elements of spontaneous activity in zebrafish Purkinje neurons, namely, sodium spikes, calcium spikes and CF-EPSPs are all present within a day after they differentiate (*Figure 9*). These three properties seem to persist at least until late larval stages. Further, bistability also appears soon after Purkinje neuron specification and persists at least until 19 dpf, indicating that Purkinje neuron electrical properties mature rapidly. However, fewer CF EPSCs were seen at 19 dpf

compared to 4 dpf. While this could reflect a developmental decrease in the firing rates of olivary neurons, there may also be a gradual maturation of CF inputs from many relatively weaker inputs at 4 dpf to few strong inputs at 19 dpf. In adult vertebrates, each Purkinje neuron is innervated by a single CF. Analogous to the neuromuscular junction, CF to Purkinje neuron synapses progress from polyinnervation to single innervation through a process of synapse elimination. In rodents, this process occurs during the first two weeks of post-natal life (*Hashimoto and Kano, 2005*). It is not yet clear when the process of synapse elimination is complete in zebrafish but our data suggest a gradual decrease in the number of CF synapses onto single Purkinje neurons between 4 dpf and 19 dpf. Further morphological and physiological experiments are required to determine the earliest stage at which Purkinje neurons are innervated by single CF.

### Functional relevance of bistability in Purkinje neurons

We have shown that Purkinje neuron bursts are highly correlated with motor neuronal spikes. Our data reveal that initiation of motor episodes is accompanied by increase in CF inputs that depolarize the Purkinje neurons (*Figure 8*). Such depolarization triggers bursts, presumably by recruiting persistent sodium, high threshold calcium and calcium-dependent potassium conductances and/or due to co-incident PF inputs. Though we did not find a tonic cell during our simultaneous motor episode recordings, we propose that mode switching could serve as a 'clutch' that engages or disengages Purkinje neuron activity to locomotor behavior. Although cells in the tonic mode might continue to receive CF inputs during motor episodes, their simple spike rates may not be modulated to the same extent as those cells in the 'down' state. By mode switching from tonic to bursting, Purkinje neurons might choose to 'listen in' on locomotion related neural inputs.

We also observed heterogeneity in the timing of Purkinje neuron depolarization with respect to motor episode initiation. The time to peak depolarization was different for different cells (*Figure 8E*). This leads us to propose that Purkinje neurons might use CF-driven bursting to generate a distributed representation of motor events. It remains to be seen whether these CF-EPSPs play a role during motor learning. With the ability to record from Purkinje neurons intracellularly during fictive motor behavior, experiments to address the above question are within reach.

## Materials and methods

### Experimental animals and microinjection

All experiments were approved by the institutional animal ethics committee and the institutional biosafety committee. Indian wild type zebrafish (*Danio rerio*) of mixed sex at the respective stages were used for all experiments. Zebrafish embryos at the 1–2 cell stage were injected with *aldoca:gap43-Venus* (gift from Prof. Masahiko Hibi, Nagoya University, Japan) and Tol2 transposase mRNA (Tol2 constructs were a gift from Prof. Koichi Kawakami, National Institute of Genetics, Japan). Embryos were grown in E3 medium (composition in mM: 5 NaCl, 0.17 KCl, 0.33 CaCl$_2$, and 0.33 MgSO$_4$, pH 7.8) and then screened for Venus expression at 4 dpf. Larvae showing mosaic labeling of Purkinje neurons with Venus were used for experiments at 4, 7 and 19 dpf.

### Electrophysiology

Loose patch and whole cell patch clamp recordings were done in wild type larvae at different stages at room temperature (23–24°C). The recording set up is shown in *Figure 1B*. The larvae were anesthetized in 0.01% MS222 and pinned onto a Sylgard (Dow Corning, Midland, MI, United States) dish using fine tungsten wire (California Fine Wire, Grover Beach, CA, United States). The MS222 was then replaced with external saline (composition in mM; 134 NaCl, 2.9 KCl, 1.2 MgCl$_2$, 10 HEPES, 10 Glucose, 2.1 CaCl$_2$, 0.01 D- tubocurare; pH: 7.8; 290 mOsm) and the skin over the head was carefully peeled off to expose the brain. The recording chamber was then transferred to the rig apparatus. All recordings were done in an awake, in vivo condition. Purkinje neurons only in the CCe (*Figure 1C*) were recorded from. The cells were observed using a 60× water immersion objective on a compound microscope (Olympus BX61WI). Cells showing Venus expression in the cerebellum were targeted for recordings.

Loose patch recordings were done on Purkinje neurons using pipettes filled with external saline that were 6–7 MΩ in resistance. After establishing the seal (20 MΩ to 2 GΩ), recordings were obtained at I = 0 mode such that no external current stimulus was applied. Recordings were acquired with

Multiclamp 700B amplifier, Digidata 1440A digitizer and pCLAMP software (Molecular Devices, Sunnyvale, CA, United States). The data was low pass filtered at 2 kHz and sampled at 20 kHz with a gain of 100.

For whole cell patch clamp recordings, pipettes of tip diameter 1–1.5 μm and resistance of 12–16 MΩ were pulled with thick walled borosilicate capillaries (1.5 mm OD; 0.86 mm ID; Warner Instruments, Hamden, CT, United States) using a Flaming Brown P-97 pipette puller (Sutter Instruments, Novato, CA, United States). Two types of pipette internal solution were used. A Potassium gluconate based solution (composition in mM: 115 K gluconate, 15 KCl, 2 MgCl$_2$, 10 HEPES, 10 EGTA, 4 MgATP; pH: 7.2; 290 mOsm) was used for most current clamp and voltage clamp recordings. A Cesium gluconate based solution (composition in mM: 115 Cs hydroxide, 115 Gluconic acid, 15 CsCl, 2 MgCl$_2$, 10 HEPES, 10 EGTA, 4 MgATP) was used to record calcium currents in voltage clamp mode. The pipette internal solution always contained 30 μg/ml of Sulphorhodamine dye (Sigma-Aldrich, St.Louis, MO, United States) for visualisation of the patched cell morphology. Colocalization of the sulphorhodamine volume fill (red; *Figure 1E*) with membrane tagged Venus (green; *Figure 1D*) was used as a reporter for successful targeted recordings (*Figure 1F*).

A total of 9 cells at 4 dpf, 78 cells at 7 dpf, and 10 cells at 19 dpf were recorded for this work. Of these a small minority were cells not labeled by Venus (5 cells at 4 dpf, 18 cells at 7 dpf and 6 cells at 19 dpf), but showed other Purkinje neuronal features such as spiny dendrites, caudal or dorsal dendrites with respect to the cell body and large tear-drop shaped cell bodies. Their input resistance and whole cell capacitance were within range of those seen in Venus-expressing Purkinje neurons (see 'Results' section).

Whole cell recordings were acquired using Multiclamp 700B amplifier, Digidata 1440A digitizer and pCLAMP software (Molecular Devices). The data was low pass filtered at 2 kHz using a Bessel filter and sampled at 20 kHz at a gain of 1. Membrane potentials mentioned were not corrected for liquid junction potential which was measured to be +8 mV for the potassum gluconate based internal solution.

The following drugs were perfused in the bath at concentrations mentioned, whenever required. TTX (1 μM, Alomone labs, Israel), Bicuculline methiodide (10 μM, Sigma-Aldrich), APV (40 μM, Tocris Bioscience, United Kingdom), CNQX (20 μM, Tocris), Gabazine (10 μM, Sigma-Aldrich) and Cadmium chloride monohydrate (200 μM, Loba Chemie, India). QX314 (5 mM, Alomone labs) was included in the patch pipette in some experiments to block sodium channels intracellularly.

To stimulate the CFs, a bipolar electrode (FHC, Bowdoin, ME, United States) was placed at the lateral edge of the cerebellum at the level of the ganglionic layer, where the olivary fibers are known to enter the cerebellum (*Takeuchi et al., 2015*). A single pulse of 2 ms and ranging in amplitude from 50 μA to 800 μA was applied using an Isoflex stimulus isolator (AMPI, Israel) driven by pClamp (Molecular Devices).

Fictive motor patterns were recorded as described in *Thirumalai and Cline (2008)*. Briefly, the larva was anesthetized in 0.01% MS222 and pinned onto a piece of Sylgard. The MS222 was then replaced with external saline and the skin on the tail was peeled to expose the underlying myotomes. Pipettes were pulled from thin-walled borosilicate glass (1.5 mm OD and 1.1 mm ID, Sutter Instruments), filled with external saline and had resistances in the range of 0.7–1.2 MΩ. The pipette was placed at a myotomal boundary and mild suction applied to get motor nerves into the suction electrode. Signals were amplified and digitized as above.

## Analysis

Neurons with a resting membrane potential higher than −40 mV and whose series resistance was more than 10% of the input resistance were excluded from the data set. Series resitance was monitored at the beginning and end of recordings and did not change by more than 10% in any neurons in our dataset. Events were detected offline using threshold based search algorithms in Clampfit (Molecular Devices). Graphs were plotted using Microsoft Excel and MATLAB (The Mathworks, Natick, MA, United States). For each experiment, the number of cells recorded from (N) is indicated in the respective figure legend. Statistical comparisons were performed in MATLAB. Significance was tested using the appropriate test (Mann–Whitney, Chi-square or Kruskal Wallis) as indicated in the results.

## Acknowledgements

We thank Prof. Masahiko Hibi, Nagoya University, Japan, for the aldoca:gap43-Venus construct; Prof. Koichi Kawakami, National Institute of Genetics, Japan for the Tol2 transposase construct. We also

wish to thank Mr Sriram Narayanan for technical assistance and Mr Manjunath for maintaining our fish facility. This project was funded by CSIR-NET fellowship (MS), Wellcome Trust DBT India Alliance Intermediate Fellowship (VT), DBT research grant (VT) and NCBS Plan funds (VT).

## Additional information

### Funding

| Funder | Grant reference | Author |
|---|---|---|
| Wellcome Trust DBT India Alliance | 500040-Z-09-Z | Vatsala Thirumalai |
| Department of Biotechnology, Ministry of Science and Technology (DBT) | BT/PR4983/MED/30/790/2012 | Vatsala Thirumalai |
| Council of Scientific and Industrial Research (CSIR) | 09/860(0142)/2012-EMR-I | Mohini Sengupta |
| Department of Atomic Energy, Government of India (DAE) | | Vatsala Thirumalai |

The funders had no role in study design, data collection and interpretation, or the decision to submit the work for publication.

### Author contributions

MS, Conception and design, Acquisition of data, Analysis and interpretation of data, Drafting or revising the article; VT, Conception and design, Analysis and interpretation of data, Drafting or revising the article

### Ethics

Animal experimentation: All experiments described in this study were approved by the Institutional Animal Ethics Committee (IAEC; approval number VT-01/12/2010), and the Institutional Biosafety Committee (IBSC; approval number 14 IBSC/29.4.11), National Centre for Biological Sciences, Bangalore.

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
