## [Decision Letter]

Thank you for submitting your work entitled “AMPA receptor mediated synaptic excitation drives state-dependent bursting in Purkinje neurons of zebrafish larvae” for peer review at *eLife*. Your submission has been favorably evaluated by Timothy Behrens (Senior Editor) and three reviewers, one of whom is a member of our Board of Reviewing Editors.

The following individuals responsible for the peer review of your submission have agreed to reveal their identity: Ronald L. Calabrese (Reviewing Editor and peer reviewer) and Jan-Marino Ramirez (peer reviewer).

The reviewers have discussed the reviews with one another and the Reviewing Editor has drafted this decision to help you prepare a revised submission.

The authors present an electrophysiological analysis of identified Purkinje neurons in awake zebrafish larvae. This analysis shows that these neurons exist in two alternative states (bistability), bursting and tonic spiking, and that they can toggle between them either with artificial current injection or mediated through AMPAR EPSPs from Climbing Fibers (CF). They also show that toggling to the bursting state occurs at the onset of fictive locomotion via the CF input. These experiments answer a long-standing question of whether PC bistability occurs in vivo and the mechanisms of toggling.

The consensus is that these experiments are carefully done, and the data are solid and interesting. However, there are several detailed suggestions by Reviewer #2, and all the minor comments of Reviewer #1 should be addressed. In consultations among the reviewers a few other concerns arose that require revision:

1) The presentation, especially in the Discussion, minimizes the impact by emphasizing similarities to mammals, where similar studies have been performed. The cerebellum in zebrafish is organized very differently – much more complex in fact. While the Introduction does a good job pointing out these differences, the Discussion seeks to generalize and it is unwise to generalize too far. We recommend that you revise the discussion of your results to give it an evolutionary spin and discuss the complexity of the zebrafish in relation to what is left in the mammal, and how the bistability is perhaps preserved as one aspect. The generic discussion of “we use the zebrafish to study bistability because it is a simpler system” minimizes impact because a lot can also be done in primates.

2) The authors have shown a correlation between motor bursts and bistability that is too strongly emphasized. Similar correlations have been observed in primates. The motor burst in question is fictive and not behavioral, and the precise timing is not established. Revision is needed to bring the results into perspective.

Reviewer #1 (Minor comments):

1) In the subsection “Narrow spikes are sodium action potentials”, do these spikes correspond to the narrow spikes in control conditions? Are the large amplitude events eliminated?

2) In Table 1, give N's for all entries. Be consistent in the use of N and n throughout.

3) In Figure 5, are regression lines plotted? When you say that inter-event interval did not vary with membrane potential, how was this determined statistically? Correlation? Regression slopes?

4) In the subsection “Ontogeny of Purkinje neuron spontaneous activity patterns”, the argument is weak because the reduction in EPSP frequency could be due to a reduction in CF firing rate. The argument must be made subsequently by comparing the results to known development events in other systems.

5) In Figure 5—figure supplement 2, please explain the plots in panel B in the legend.

Reviewer #2:

This interesting manuscript characterizes bistability in Purkinje cells using the zebrafish larvae as a model system. This model allows in vivo patch clamp recordings while the animal generates fictive movements. The use of an in vivo model is the major strength of this study, and the data obtained in this larva are reminiscent to observations made in primates – i.e. that the onset of bistability correlates with the onset of motor bursts. The data are well presented, and the Discussion is interesting for a general readership. The significance of this study is discussed in the context of mammalian neurobiology, and the fact that the findings are in part consistent with insights gained in mammals is reassuring that the zebrafish cerebellum, despite all its major anatomical differences may have similar physiological roles. Nevertheless, it is important to keep in mind that the zebrafish larva is not a mammal, and it is certainly not an adult mammal. Thus, it is a bit of a stretch to assume that bistability is a general principle of cerebellar function based on this study. This does not deter from the fact that the data are interesting, and that the experiments are well done. As for the data I have only a few comments:

1) The authors state, at the beginning of the Discussion, that they have demonstrated that Purkinje cells show bistability in a behaving animal. This is clearly overselling the data. The authors studied bistability in relation to fictive motor bursts – which is not exactly a well-defined behavior, in particular since the activity is also fictive. While it may likely represent the onset of locomotion, the authors need to be a bit more careful. At the same time, I don't want to question the fact that there is a relationship between bistability and motor activation, but stating “behaving animal” is going too far – even for someone who is a strong advocate of the analysis of fictive activity.

2) The figures should also show control conditions when a drug is tested. This is for example the case in Figure 3 and Figure 4 (voltage recordings are not considered as a proper control for current recordings) as well as Figure 6.

3) Figure 6 shows the switch from a tonic to a bursting mode. To me it would be more convincing to also see when the bursting switches again back to the tonic mode. The Figure A could also be interpreted as a recording artifact: i.e. it could represent a recording that is initially bad (leaky) and then settles into a normal membrane potential. Specially concerning is that the transition seems to be gradual.

4) CF responses: The data show spontaneous and evoked CF responses, which are convincing. The authors further propose that these responses are mediated by AMPA type receptors, and that CF input triggers burst spiking in Purkinje cells. All these assumptions could be true. However, there seems to be one data point that is inconsistent with this claim: spontaneous CF responses disappear (Figure 5), while evoked CF responses are still present (Figure 7) under CNQX. Please address this issue.

5) Figure 8 is interesting, but it seems to me that the motor neuron fires before Purkinje cell bursts. In addition, based on the current knowledge, Purkinje cell bursts will shut down the efferent cells or eurydendroid cells, which project to pre- or motor neurons. Thus, the authors need to discuss in more detail how they believe Purkinje cell bursts turn on motor neuron activity.

6) Miniature signals should be excluded when trying to demonstrate under voltage-clamp conditions synaptic responses without stimulation, such as done in Figure 5—figure supplement 2.

Reviewer #3:

This study by Sengupta and Thirumalai examines the intrinsic and synaptic properties of Purkinje neurons in larval zebrafish. The results show that Purkinje neurons display two modes of activity consisting of tonic and burst firing that co-exist in the same neurons. Further, it is shown that burst firing can be induced by synaptic inputs from climbing fibers. The results are straightforward and in many aspects similar to those reported in other fish and mammals.

I find this study easy to read and the results are clear and well illustrated. Its strength is that it is done in larval zebrafish where Purkinje neuron activity can be correlated with motor behavior. From a comparative point of view, this study shows that the main properties of Purkinje neurons are conserved between fish and mammals. In my opinion, however, this study does not represent a significant step forward.

Reviewer #3 (Minor comments):

1) The existence of efference copy from the spinal cord to cerebellum is well documented in mammals. Indeed, it is known that cerebellum “listens in” motor commands as well as sensory feedback. The authors should cite some of these references.

2) In Figure 8, what do the blue and green arrowheads represent?

---

## [Author Response]

*1) The presentation, especially in the Discussion, minimizes the impact by emphasizing similarities to mammals, where similar studies have been performed. The cerebellum in zebrafish is organized very differently – much more complex in fact. While the Introduction does a good job pointing out these differences, the Discussion seeks to generalize and it is unwise to generalize too far. We recommend that you revise the discussion of your results to give it an evolutionary spin and discuss the complexity of the zebrafish in relation to what is left in the mammal, and how the bistability is perhaps preserved as one aspect. The generic discussion of “we use the zebrafish to study bistability because it is a simpler system” minimizes impact because a lot can also be done in primates*.

We have rewritten the Discussion, leaving out parts that were generalizing too far and adding in parts that place these results in an evolutionary context. The Discussion now points to differences and similarities between fish and mammal and places the bistability of Purkinje neurons in this context.

*2) The authors have shown a correlation between motor bursts and bistability that is too strongly emphasized. Similar correlations have been observed in primates. The motor burst in question is fictive and not behavioral, and the precise timing is not established. Revision is needed to bring the results into perspective*.

The correlations observed in primates are with actual movement and not with motor neuronal spikes. Movements occur on a much slower time scale while correlations with motor neuronal spike times can be more temporally precise. Previous studies in vivo using extracellular recordings of Purkinje neuron simple spikes in rodents, cats and monkeys have shown variability in Purkinje neuron spike times with respect to movement (Armstrong and Edgley, 1984; [18]; Sauerbrei et al., 2015). But it has been unclear whether the spike time variability reflects kinematic variability or variability in external forces or the variability is entirely due to the intrinsic dynamics of the underlying circuit.

By recording motor neuronal spikes instead of movement, we have eliminated kinematic variability and variability in external forces. Our data show that even when correlated with motor neuronal spikes, Purkinje neurons show jitter in their burst timing (see below). This is a novel result.

We agree that the motor episodes we record are fictive and not actual behavior. We have removed references to ‘behaving animal’ wherever present and have replaced it with ‘fictive motor output’. We have also included additional analyses of the data shown in Figure 8 to bring out the relationship between Purkinje neuron bursts and motor neuron spikes. These analyses show that significantly more CF-EPSPs arrive within 1s after motor episode initiation and that Purkinje neurons show significant depolarization in this window. Trial-to-trial, we see jitter spread over tens of milliseconds in the timing of the Purkinje neuron burst with respect to motor episode initiation. There is also cell-to-cell variability in the timing of the burst with respect to motor episode initiation suggesting that Purkinje neurons might differentially represent timing of motor events by varying the timing of their burst.

Reviewer #1 (Minor comments):

1) In the subsection “Narrow spikes are sodium action potentials”, do these spikes correspond to the narrow spikes in control conditions? Are the large amplitude events eliminated?

Yes, these spikes correspond to narrow spikes. We have included a figure to show that these narrow spikes are abolished by TTX (Figure 3—figure supplement 2). Yes, the large amplitude events are eliminated and we have mentioned this in the Results section (in the subsection “Broad spikes are dependent on voltage-dependent calcium channel activation”).

*2) In*
Table 1*, give N's for all entries. Be consistent in the use of N and n throughout.*

Numbers of cells and events have been listed in Table 1. ‘N’ is used throughout.

*3) In*
Figure 5*, are regression lines plotted? When you say that inter-event interval did not vary with membrane potential, how was this determined statistically? Correlation? Regression slopes?*

Correlation coefficients and their P values were calculated and have been included in the Results section (in the subsection “Large amplitude events are putative climbing fiber EPSPs”).

4) In the subsection “Ontogeny of Purkinje neuron spontaneous activity patterns”, the argument is weak because the reduction in EPSP frequency could be due to a reduction in CF firing rate. The argument must be made subsequently by comparing the results to known development events in other systems.

We agree with the reviewer. These lines have now been modified appropriately.

*5) In*
Figure 5—figure supplement 2*, please explain the plots in panel B in the legend.*

Explanation for box plots are now included in the legend for Figure 5—figure supplement 2.

Reviewer #2:

*1) The authors state, at the beginning of the Discussion, that they have demonstrated that Purkinje cells show bistability in a behaving animal. This is clearly overselling the data. The authors studied bistability in relation to fictive motor bursts – which is not exactly a well-defined behavior, in particular since the activity is also fictive. While it may likely represent the onset of locomotion, the authors need to be a bit more careful. At the same time, I don't want to question the fact that there is a relationship between bistability and motor activation, but stating “behaving animal” is going too far – even for someone who is a strong advocate of the analysis of fictive activity*.

We have rewritten this sentence incorporating the reviewer’s suggestion (in the subsection “Larval zebrafish as a model system to study the cerebellum and motor learning”).

*2) The figures should also show control conditions when a drug is tested. This is for example the case in*
Figure 3
*and*
Figure 4
*(voltage recordings are not considered as a proper control for current recordings) as well as*
Figure 6*.*

Control traces have been inserted in Figures 3, 4 and 6.

*3)*
Figure 6
*shows the switch from a tonic to a bursting mode. To me it would be more convincing to also see when the bursting switches again back to the tonic mode. The Figure A could also be interpreted as a recording artifact: i.e. it could represent a recording that is initially bad (leaky) and then settles into a normal membrane potential. Specially concerning is that the transition seems to be gradual*.

We agree that a slow hyperpolarization can be interpreted as gradual improvement of the recording quality. However, we monitor access resistance and input resistance before and after each recording and include only those cells where the recording is stable. The trace that was shown in Figure 6 is one such. Also, we see spontaneous switching from tonic to bursting, and bursting to tonic. In any case, to avoid any confusion, we now show a spontaneous switch from bursting to tonic mode mediated by rapid depolarization in Figure 6. This cell ultimately returned to bursting state after tens of seconds. These results are discussed in the subsection “The same neuron can toggle between bursting and tonic states”.

*4) CF responses: The data show spontaneous and evoked CF responses, which are convincing. The authors further propose that these responses are mediated by AMPA type receptors, and that CF input triggers burst spiking in Purkinje cells. All these assumptions could be true. However, there seems to be one data point that is inconsistent with this claim: spontaneous CF responses disappear (*Figure 5*), while evoked CF responses are still present (*Figure 7*) under CNQX. Please address this issue*.

CNQX abolishes spontaneous and evoked CF responses. The large amplitude events marked by grey arrowheads in Figure 7 are stimulus artifacts. The small amplitude spikelets seen between the stimulus artifacts are not CF EPSPs. They appear to be filtered sodium spikes appearing well below threshold and may be spikes originating in electrically coupled neurons. Electrical synapses are prevalent in the Purkinje cell layer at this stage (Jabeen and Thirumalai, 2013).

*5)*
Figure 8
*is interesting, but it seems to me that the motor neuron fires before Purkinje cell bursts. In addition, based on the current knowledge, Purkinje cell bursts will shut down the efferent cells or eurydendroid cells, which project to pre- or motor neurons. Thus, the authors need to discuss in more detail how they believe Purkinje cell bursts turn on motor neuron activity*.

We have now included detailed analyses of these data and discuss them in the Results section (in the subsection “Bursts are triggered during motor episodes”). The timing of the Purkinje neuron burst with respect to the start of the motor episode is variable from trial-to-trial and from cell-to-cell. However, averages over many trials suggest that each Purkinje neuron bursts within a preferred window that is different for each of the cells recorded from. These data suggest that Purkinje neurons as a population may have a distributed representation of the time of initiation of motor episodes.

We do not at all think that Purkinje cell bursts turn on motor activity. The bursts probably represent an internal image of expected motor output, which is constructed from an efference copy of the motor episode. Further experiments are required to verify this aspect.

Purkinje neurons are GABAergic, and the reviewer is correct in pointing out that the net effect of Purkinje neuron spikes is to shut down the efferent cells. However, earlier studies suggest that synchronized inhibition from Purkinje neurons could result in precisely timed spikes in the efferent cells (Person and Raman, 2012). However, this is beyond the scope of our manuscript and therefore we have refrained from speculating about downstream effects of Purkinje neuron bursts.

*6) Miniature signals should be excluded when trying to demonstrate under voltage-clamp conditions synaptic responses without stimulation, such as done in*
Figure 5—figure supplement 2.

The small synaptic currents seen in Figure supplement 5B (now labeled Figure 5—figure supplement 2) panel A are for the most part not miniature synaptic currents. We have recorded mEPSCs from Purkinje neurons at 7dpf. mEPSCs in Purkinje neurons are infrequent with a median inter event interval of 5s as opposed to a median inter-event interval of 1.8s for the PF synaptic inputs shown in this figure. Based on this, we estimate that more than 67% of the events reported in this figure are non-miniature synaptic events. In any case, the recordings shown in this figure were performed with a potassium gluconate patch internal solution. When recording mEPSCs we use a cesium gluconate patch internal solution, as minis are usually not visible at the soma if leak currents are not blocked with cesium.

Reviewer #3:

*This study by Sengupta and Thirumalai examines the intrinsic and synaptic properties of Purkinje neurons in larval zebrafish. The results show that Purkinje neurons display two modes of activity consisting of tonic and burst firing that co-exist in the same neurons. Further, it is shown that burst firing can be induced by synaptic inputs from climbing fibers. The results are straightforward and in many aspects similar to those reported in other fish and mammals*.

*I find this study easy to read and the results are clear and well illustrated. Its strength is that it is done in larval zebrafish where Purkinje neuron activity can be correlated with motor behavior. From a comparative point of view, this study shows that the main properties of Purkinje neurons are conserved between fish and mammals. In my opinion, however, this study does not represent a significant step forward*.

We thank the reviewer for his/her positive comments but we respectfully disagree with him/her regarding the significance of this manuscript. This study for the first time unequivocally demonstrates that bistability is an inherent property of Purkinje neurons in vivo and that the bistability is functionally important during motor episodes. Prior to this study, the existence of bistability in vivo in Purkinje neurons had been debated. As the reviewer correctly points out, the use of the larval zebrafish preparation has allowed us to record intracellularly in the absence of anesthesia and this has allowed us to address this long-standing controversy. Given the conservation of several other properties of Purkinje neurons between fish and mammals, and in light of earlier in vitro experiments in mammals where bistability was observed, the proof for bistability in zebrafish Purkinje neurons in vivo can be taken as proof of its existence in mammals in vivo. This finding in itself is significant for the broader neuroscience research community.

Secondly, we now show that burst timing in Purkinje neurons is variable even when correlated with motor neuronal spikes. Again, the use of the larval zebrafish preparation allowed us to record intracellularly from Purkinje neurons and extracellularly from motor neurons simultaneously. Exploiting this advantage, we discovered that in a 1s window after the initiation of motor episodes, the incidence of CF-EPSPs is significantly higher and that this leads to membrane depolarization and bursting. However, the exact timing of the Purkinje neuron bursts is variable. This again is a significant finding as it suggests an intrinsic circuit-based mechanism for distributed representations of motor episode timing among Purkinje neurons.

Reviewer #3 (Minor comments):

*1) The existence of efference copy from the spinal cord to cerebellum is well documented in mammals. Indeed, it is known that cerebellum “listens in” motor commands as well as sensory feedback. The authors should cite some of these references*.

We observe CF-EPSPs and Purkinje neurons bursts that are timed to occur within 1 s from initiation of motor episodes. However, since the preparation is paralyzed, no actual movement occurs and therefore there is no proprioceptive feedback signal. We are not sure whether the efference copy that triggers Purkinje neuron activity originates in the spinal cord or elsewhere. In any case, we have cited a few papers that discuss motor representation in Purkinje neurons (39; 27; 3; 18).

*2) In*
Figure 8*, what do the blue and green arrowheads represent?*

The blue arrowhead represents the initiation of Purkinje neuron burst and the green arrowhead represents initiation of motor neuron spiking. These are mentioned in the figure legend now.